# Current Concept of Quantitative Sensory Testing and Pressure Pain Threshold in Neck/Shoulder and Low Back Pain

**DOI:** 10.3390/healthcare10081485

**Published:** 2022-08-07

**Authors:** Hidenori Suzuki, Shu Tahara, Mao Mitsuda, Hironori Izumi, Satoshi Ikeda, Kazushige Seki, Norihiro Nishida, Masahiro Funaba, Yasuaki Imajo, Kiminori Yukata, Takashi Sakai

**Affiliations:** 1Department of Orthopaedics Surgery, Graduate School of Medicine, Yamaguchi University, Yamaguchi 755-8505, Japan; 2Pain Management Research Institute, Yamaguchi University Hospital, Yamaguchi 755-8505, Japan; 3Department of Rehabilitation, Yamaguchi University Hospital, Yamaguchi 755-8505, Japan

**Keywords:** quantitative sensory testing, pressure pain threshold, musculoskeletal pain, reference value, low back pain, neck/shoulder pain

## Abstract

In recent years, several published articles have shown that quantitative sensory testing (QST) and pressure pain threshold (PPT) are useful in the analysis of neck/shoulder and low back pain. A valid reference for normal PPT values might be helpful for the clinical diagnosis of abnormal tenderness or muscle pain. However, there have been no reliable references for PPT values of neck/shoulder and back pain because the data vary depending on the devices used, the measurement units, and the area examined. In this article, we review previously published PPT articles on neck/shoulder and low back pain, discuss the measurement properties of PPT, and summarize the current data on PPT values in patients with chronic pain and healthy volunteers. We also reveal previous issues related to PPT evaluation and discuss the future of PPT assessment for widespread use in general clinics. We outline QST and PPT measurements and what kinds of perceptions can be quantified with the PPT. Ninety-seven articles were selected in the present review, in which we focused on the normative values and abnormal values in volunteers/patients with neck/shoulder and low back pain. We conducted our search of articles using PubMed and Medline, a medical database. We used a combination of “Pressure pain threshold” and “Neck shoulder pain” or “Back pain” as search terms and searched articles from 1 January 2000 to 1 June 2022. From the data extracted, we revealed the PPT values in healthy control subjects and patients with neck/shoulder and low back pain. This database could serve as a benchmark for future research with pressure algometers for the wide use of PPT assessment in clinics.

## 1. Introduction

Musculoskeletal disease is a worldwide problem for which healthcare assistance is frequently sought. Low back pain (LBP) and neck/shoulder pain are the most common musculoskeletal conditions that evolve into chronic problems [1,2]. Musculoskeletal pathology may initiate chronic pain, but the pain is often also modulated by sensory inputs from the peripheral and central nervous systems [3]. Central sensitization is involved in the chronification of pain, which manifests as hypersensitivity to pain and is spread beyond the areas immediately affected by musculoskeletal pathology [4]. It continues to be challenging to detect and measure hypersensitivity in clinical practice, and no consensus has been reached on which tools are best for assessing musculoskeletal pain [2].

Quantitative sensory testing (QST) combines simple tools that can assess the ability to perceive touch, vibration, proprioception, and sensitivity to pinpricks or blunt pressure and to cold or heat stimuli [2]. QST and the assessment of the pressure pain threshold (PPT) have become commonplace in clinical neurophysiology units [5,6,7,8]. QST/PPT uses psychophysical tests defined as stimuli with predetermined physical properties based on specific measurement protocols for the analysis of somatosensory aberrations. QST/PPT measures sensory stimuli and can be used to assess somatosensory system functions, the measurement of altered peripheral and/or central pain sensitivity, and descending pain modulation [8].

PPT is the QST parameter most frequently used to investigate local and widespread hyperalgesia. PPT reflects sensitivity to pain and can be measured by either electronic or mechanical pressure algometry. In this test, subjects report when gradually applied pressure changes from a feeling of pressure to that of pressure combined with pain [8]. The advantages of PPT include its simplicity and rapid measurement time compared to other QST protocols in which measurement time is longer and requires more effort [5,9,10].

Several articles published over recent years have shown the usefulness of PPT in analyzing neck/shoulder and back pain [11,12,13,14,15,16,17]. Evidence from these studies indicates that PPT would appear to be a useful tool for analyzing the pathogenesis, classification, differential diagnosis, and prognosis of neck/shoulder and back pain [11]. However, the assessment of neck/shoulder and back pain with PPT has one main problem. Standardized normative values for neck/shoulder and back pain conditions are lacking and need to be developed. Although valid reference values indicative of a normal PPT would aid in the clinical diagnosis of muscle pain or abnormal tenderness, no such reliable values currently exist for neck/shoulder and back pain [17].

Therefore, the aims of the present article are to review previously published articles on PPT for neck/shoulder pain and LBP, to discuss measurement properties of PPT, and to review and summarize the present data on PPT values in patients with chronic neck/shoulder and LBP and healthy volunteers, based on our search of the current knowledge base on PPT. We also reveal previous issues related to the PPT evaluation of patients with chronic pain and discuss the future of PPT for widespread use in general clinics.

## 2. Quantitative Sensory Testing

QST collectively refers to a group of procedures that assess the perceptual response to systematically applied and quantitative sensory stimuli to characterize somatosensory function or dysfunction [8,18]. QST involves procedures that test perception, pain threshold, and pain tolerance thresholds for different stimuli based on the application of standardized pressure, vibration, thermal, or electrical impulses. QST measures the response to sensory stimuli and can be used to assess somatosensory system function, the measurement of altered peripheral and/or central pain sensitivity, and descending pain modulation [8,19].

By selecting various QST modalities, different fibers can be tested. The function of Aδ fibers is represented by the cold detection threshold, that of C fibers by the heat detection threshold, that of nociceptive C fibers mainly by the heat pain threshold, and that of Aβ fibers by mechanical detection and vibration [11,18,20]. The thermal, mechanical, and electrical tests commonly applied in QST are listed in Table 1 [20,21,22].

## 3. Pressure Pain Threshold

Among the QST parameters, PPT is the most frequently assessed. PPT is determined by applying a mechanical stimulus to determine the moment that the stimulus-induced sensation of pressure first changes to that of pain [23]. This allows the quantification of the PPTs of skin and muscle. An algometer is often used to apply pressure to sites both close and far from the location of the subject’s pain. Factors such as sex, the investigator, and the apparatus used may affect the measurement of PPT by pressure algometry. The reliability of PPT based on raters or measurement frequencies is reported to be relatively high [23,24].

### 3.1. Perceptions of Peripheral and Central Sensitization Can Be Quantified by PPT

PPT can be used to evaluate peripheral and central sensitization. Tenderness experienced with blunt pressure may be caused by the peripheral sensitization of primary afferents or central sensitization [25]. Because PPT preferentially activates deep afferents, it is a good clinical device for measuring peripheral sensitization. Hyperalgesia of the affected area to blunt mechanical stimuli is thought to reflect the peripheral sensitization of Aδ and C fibers. Unlike cutaneous nociceptors, which are particularly sensitive to thermal stimuli, nociceptors in deep somatic tissue, such as joints and muscles, exhibit a pronounced sensitivity to mechanical stimuli [25,26].

PPT can also assess central sensitization, which can cause mechanical receptive fields to expand. Although this might account for some local spreading of tenderness, the alteration of pathways descending from the brainstem is more likely to result in widespread or generalized tenderness. A widespread lowering of PPT may reflect the dysfunction of the endogenous pain inhibitory mechanism [25,26,27].

### 3.2. PPT Analysis in Neck/Shoulder and Low Back Pain

PPT is also effective in disorders involving musculoskeletal pain. Pressure stimuli generated by an algometer can target muscles or fascia, thus indicating that the application of such stimuli would be suitable for patients with muscle or joint pain [28]. The reliability of algometer use in patients with musculoskeletal pain has been established. In addition, in a reliability study using several PPTs, algometers were reported to have the least variability and highest reliability in assessing musculoskeletal pain [26,28]. Distinguishing between alterations in peripheral and central pain processing in patients with musculoskeletal pain is important, as central sensitization is considered a potential influence in the development and maintenance of chronic pain. There are many reports on the use of PPT in patients with neck/shoulder, low back, and other musculoskeletal pain [28]. Furthermore, PPT might be valuable in predicting postoperative pain after surgery on musculoskeletal structures [2,6,9]. PPT can be used to evaluate the pathophysiology of peripheral and central sensitization in patients with neck/shoulder and LBP and is useful when analyzing the pathogenesis of chronic pain as well as its classification, differential diagnosis, and prediction [12].

They are meaningful evaluation techniques for patients with chronic pain because it is generally very difficult to objectively score how much pain the patients feel in the neck/shoulder and back area, including central sensitizations. PPT examination is one of the solutions to examining patients with chronic pain for digitalization [12]. However, no reviews of previous articles on PPT analysis for chronic neck/shoulder and LBP have been published, nor have standardized methods of assessing PPT for musculoskeletal pain been reported. Moreover, the results are different in the articles showing standardized, normative, and abnormal PPT values of neck/shoulder and LBP in volunteers/patients with and without chronic pain [12].

For these reasons, PPT is not a popular tool in the evaluation of patients with chronic pain even now in general clinics [11]. It is necessary to review all articles on PPT related to neck/shoulder and LBP in volunteers/patients with and without chronic pain in order to achieve the wide and general use of PPT examination. In addition, issues related to recent PPT analysis from the review of the published articles should be pointed out. In the next section, we discuss the previously published articles on PPT for neck/shoulder and LBP published from 2000 to 2022 (Table 2 and Table 3). We also reveal the normative and abnormal values for the neck, shoulder, and back, retrieved from the published articles, because these values would be the most important in helping clinicians to distinguish whether patients have abnormal pain. In addition, these data would be meaningful for clinical use in general clinics to help popularize PPT as an examination tool.

## 4. Systematic Review of PPT Values in Healthy Control Subjects and Patients with Neck/Shoulder and Low Back Pain

In this section, we review normal and abnormal PPT values in patients with neck/shoulder and LBP, PPT devices, and the area of PPT examination from the selected articles.

### 4.1. Methods of Literature Search and Inclusion Criteria

We conducted a systematic review according to PRISMA (Preferred Reporting Items for Systematic Reviews and Meta-Analyses) guidelines (https://prisma-statement.org/; accessed on 1 June 2022). Using the PubMed/MEDLINE database, we first identified relevant articles using the search terms “Pressure pain threshold” AND “Neck shoulder pain” OR “Back pain” published until 1 June 2022 from 1 January 2000. Based on this review of the article titles, we selected relevant titles related to our review. We excluded articles that were not written in the English language. These titles underwent an abstract review, after which unrelated titles were excluded. Additional relevant publications were identified and added after review of reference lists. The remaining articles underwent a full-text review. Articles without full text were excluded. Animal studies were also excluded.

The criteria for selection of the articles were: (1) PPT values as one of the main outcomes; (2) measurement site and method of assessment of PPT were described in detail; (3) the articles were written in English. For this review on musculoskeletal pain, we focused on neck/shoulder pain and back pain, for which PPT values are frequently reported in musculoskeletal pain.

### 4.2. Study Selection

We identified a total of 6523 articles through our database search. After title review and the removal of duplicates, 6275 articles were excluded, and 248 articles underwent abstract and full-text review. A total of 151 articles were excluded on full-text review. The reasons for the exclusion of studies were: (1) PPT value was not listed as one of the main outcomes; (2) The measurement site and method of assessment of PPT were not described in detail, and the articles were written in English; (3) There were no written in PPT values; (4) They were deemed unsuitable after a discussion. Ninety-seven studies met the criteria for review. The search flow diagram is displayed in Figure 1. Titles and abstracts of the studies identified by the search strategy were independently screened by two reviewers (H.S. and S.T.) to determine potentially relevant studies. Full texts of potentially relevant studies were retrieved and evaluated for eligibility by the same reviewers. Any disagreements were resolved via consensus; if consensus could not be reached, a third, independent reviewer (T.S.) resolved the dispute.

As shown in Table 2 and Table 3, 97 articles were selected in the present review, in which we focused on the normative values and abnormal values in volunteers/patients with neck/shoulder and LBP based on the values reported in the articles, because it was difficult to compare the effects of each treatment directly. The subjects ranged in age from 18 to 75 years old. The majority of the studies used pressure algometers manufactured by Somedic AB (Sweden) or Wagner Instruments (Greenwich, CT, USA).

### 4.3. Quality Assessment and Risk of Bias Assessment

Two review authors (H.S. and S.T.) independently performed the risk of bias assessment, and a third review author (Y.I.) was involved in case of disagreement. The Cochrane Back Review Group “risk of bias” tool was used. All studies had at least one serious risk of bias, with a consequent overall serious risk of bias for those studies. Critical appraisal revealed a spread in methodological quality. Common areas of bias were the lack of use of accepted diagnostic criteria for neck/shoulder/back pain and the lack of reporting of the validity and reliability of the measurement device. Furthermore, all studies had high selection bias because no study reported randomly selecting or consecutively recruiting participants.

### 4.4. Neck/Shoulder Pain

Forty-nine studies, in which the PPT of the neck–shoulder area was measured in patients with neck/shoulder pain and/or healthy volunteers, are listed in Table 2 [29,30,31,32,33,34,35,36,37,38,39,40,41,42,43,44,45,46,47,48,49,50,51,52,53,54,55,56,57,58,59,60,61,62,63,64,65,66,67,68,69,70,71,72,73,74,75]. The subjects of studies of healthy volunteers, workers, children, and young adults without pain were also reported in 10 articles, shown at the bottom of Table 2 [67,68,69,70,71,72,73,74,75]. These articles revealed the normative values of PPT in the neck–shoulder area. PPT was examined in the areas of the deltoid, trapezius, sternocleidomastoid, supraspinatus, and infraspinatus muscles, spinal processes, and C5–C6 zygapophyseal joints. Several devices and different units of measurement were used in the examination. The most common device used was a pressure algometer from Somedic AB, and the measurement unit was kPa. The normative PPT values obtained from the previous data ranged from 175 to 420 kPa at the neck–shoulder area, indicating variability in the data. In addition, several papers compared the data from healthy controls with those of the patients with chronic pain in the neck or shoulder. Nine articles showed the normative value, while data values in comparative tests ranged from 151 to 337 kPa at this area [33,34,35,40,41,47,48,49,53,55,56,57,58,59,65], again revealing variability in the data.

There were 39 studies that measured PPT in the neck–shoulder area of patients with neck/shoulder pain (Table 2) [29,30,31,32,33,34,35,36,37,38,39,40,41,42,43,44,45,46,47,48,49,50,51,52,53,54,55,56,57,58,59,60,61,62,63,64,65]. Algometers made by Somedic AB (units in kPa) and Wagner Instruments (units in kg/cm^2^) were mainly used for PPT measurements, but more than 10 different devices were used in the articles. The areas of PPT examination were the deltoid, trapezius, levator scapulae, semispinalis capitis, sternocleidomastoid, supraspinatus, and infraspinatus muscles, spinal processes, suboccipital muscles, and C5–C6 zygapophyseal joints. The trapezius muscle and levator scapulae were the most commonly evaluated areas of PPT [33,34,41,53,55,56,57,64,66,70,73,74], but the analyzed areas differed widely in the articles. The abnormal values of PPT in the trapezius muscle of the patients with chronic neck or shoulder pain were reported to vary from 151 to 411 kPa and 1.35 to 4.14 kg/cm^2^ [33,34,41,53,55,56,57,64], with varying data depending on the device and units used. From this review of previously published articles evaluating neck–shoulder pain using algometers to measure PPT, we think that it is difficult to show the abnormal and cutoff values of patients/volunteers with chronic neck or shoulder pain because the devices, measurement units, and PPT data vary widely across the different papers. However, a difference score above 25–81 kPa between the patients with neck or shoulder pain and healthy control subjects was a meaningful difference for PPT measured by Somedic AB algometers (Table 2) [33,34,41,53,55,56,57,64].

### 4.5. Low Back Pain

Forty-eight studies that measured PPT in the low back area in patients with LBP and/or healthy volunteers are listed in Table 3. The subjects of the studies were patients with LBP, healthy volunteers, workers, and patients with myofascial pain syndrome [76,77,78,79,80,81,82,83,84,85,86,87,88,89,90,91,92,93,94,95,96,97,98,99,100,101,102,103,104,105,106,107,108,109,110,111,112,113,114,115,116,117,118,119,120,121,122]. These articles revealed the normative values of PPT in the lower back area. The areas examined were the back, lumbar region, 1–3 cm lateral of the spinous processes, paravertebral muscle, paraspinal muscle, dorsal longissimus, erector spinae muscle, suprainterspinous ligaments, lumbar zygapophyseal joints, and gluteus maximus/medius. The common measurement devices were pressure algometers from Somedic AB and Wagner Instruments, and the measurement units used were mainly kPa, kgf, N/cm^2^, and kg/cm^2^ [75,77,78,79,81,83,86,89,92,93,95,96,97,98,100,105,106,107,111,112,113,114,115,117,118,119,120,121,122]. The normative value from the data ranged from 299 to 628 kPa at the back area [118,119,120,121,122]. Twenty-four articles showed comparisons of the healthy control data with the data of the patients with LBP. The normative values and data ranged from 314 to 560 kPa and 2.2 to 13.1 kg/cm^2^ at the low back area [75,88,90,91,93,96,99,102,104,105,106,107,109,111,112,113,114,115,116,117,118,119,120,121]. The abnormal PPT values in the patients with LBP were 322–451 kPa and 2.6–8.3 kg/cm^2^ at the low back muscle. A difference score above 52–184 kPa between the patients with LBP and the healthy control subjects indicated a meaningful difference in PPT as measured by the Somedic AB algometer (Table 3) [93,96,102,104,105,109,112,115,118].

## 5. Discussion

Previously published articles revealed that the individuals with higher scores in the pain-related questionnaire and with a higher score of central sensitization showed lower values on PPT; in addition, previous studies consistently demonstrated that there was a moderate to strong correlation between PPT value and disability/pain intensity [32,33,34,35,36,37,38,39,40,41,42,43,44,45,46,47,48,49,50,51,52,53,54,55,56,57]. Although valid reference values indicative of a normal and abnormal PPT would aid in the clinical diagnosis of pain, no such reliable values and no published review articles currently exist for neck/shoulder and back pain [17].

To the best of our knowledge, this is the first review article to summarize and evaluate the previously published articles on PPT in patients with neck/shoulder and LBP and to analyze the existing data on PPT values. In the general population, variations in PPT can be associated with several factors, including ethnicity, sex, age, anxiety, and physical activity [8,9,10,11,12,13,14,15,16,17]. Although data on PPT collected by a number of different researchers and the use of different algometers can also be sources of variation in PPT results, in general, the studies showed the high reliability of pressure algometry [22]. However, the evaluation of PPT in general clinics has not been a popular method for patients with pain, possibly because of the difference in results among the articles showing the normative and abnormal PPT values of neck/shoulder and back pain in volunteers/patients with and without pain [11] (Table 2 and Table 3). Despite the many articles dealing with PPT data, no work has synthesized assertive pain threshold values. Therefore, we performed this literature review to better understand the problem of measuring pain sensation.

The PPT values collected from several studies were obtained from varying parts of the body and thus may not be directly comparable [22,29,30,31,32,33,34,35,36,37,38,39,40]. However, using these values, we propose a database of PPTs that could serve as a benchmark for future research with pressure algometers on healthy subjects and subjects with some disease or pain.

Compared with healthy controls, patients experiencing chronic pain exhibit a significantly lower PPT [33,34,41,53,55,56,57,64,75,88,90,91,93,96,99,102,104,105,106,107,109,111,112,113,114,115,116,117,118,119,120,121,122]. The measurements of several PPT values were collected and critically compared for different body areas. These values verified that healthy subjects have higher PPTs than those who present with neck/shoulder and back pain. Moreover, we have found evidence that depending on whether the pain is located in the neck/shoulder or low back, the intensity of pain directly affects the pain threshold, indicating that as the pain level increases, the pain threshold decreases [33,34,41,53,55,56,57,64,75,88,90,91,93,96,99,102,104,105,106,107,109,111,112,113,114,115,116,117,118,119,120,121,122]. From the reviewed studies among patients assessed with a Somedic AB algometer, a meaningful difference was indicated by a difference score above 25–81 kPa at the deltoid muscle between patients with neck or shoulder pain and healthy control subjects, and by a difference score above 52–184 kPa at the paravertebral muscle between patients with LBP and healthy control subjects. We could not assess all of the abnormal differences for each device because the sample sizes in a number of the articles were too small to analyze statistically [35,40,48,73,91,102,109,116,120].

From these values, a database of PPTs in neck/shoulder and back pain can be generated. This database could serve as a benchmark for future research with pressure algometers. In addition, in conjunction with other physiological and biometric signals, it could be quite helpful in future work related to the measurement of pain. However, we still have several issues regarding PPT analysis. Each institute used each device and measured in a different way during PPT examination. This is the problem of PPT analysis for standardization. We think that worldwide guidelines are required for the standardization of PPT examination from now.

Finally, another guideline is required that describes the characteristics of an ideal algometer for the measurement of PPT. A portable algometer is needed for the measurement of the PPT of different body parts that is smaller, lighter, and cheaper than current devices. The validity and reliability of newly designed manual or electromechanical algometers can then be evaluated considering the PPT values collected and shown in this paper.

## 6. Conclusions

In conclusion, we revealed the normative and abnormal values for neck/shoulder and back pain conditions in PPT analysis from the previously published articles. However, the instruments, methods and areas of PPT examination were not standardized. The differences between each institution and facility remain issues of concern for PPT analysis and standardization.

## Figures and Tables

**Figure 1 healthcare-10-01485-f001:**
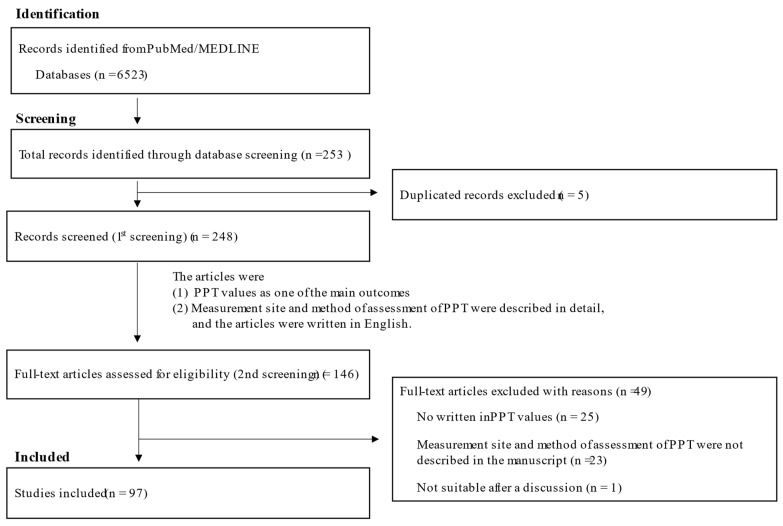
Flowchart of references—screening process in this systematic review.

**Table 1 healthcare-10-01485-t001:** Type of assessment of stimulus modalities by quantitative sensory testing (QST).

QST Type	Sensation/Modulation	Stimulus Modalities
Thermal	Warm	Warm detection threshold (WDT)
	Cold	Cold detection threshold (CDT)
	Pain	Heat pain threshold (HPT)
		Cold pain threshold (CPT)
		Suprathreshold heat pain intensity (STHPI)
Mechanical	Vibration	Vibration detection threshold (VDT)
	Pain	Pressure pain threshold (PPT)
		Suprathreshold pressure pain intensity (STPPI)
		Pressure pain tolerance (PPTol)
Electrical	Pain	Electrical pain threshold (EPT)
		Electrical pain tolerance (EPTol)
Dynamic	Wind-up	Temporal summation (TS)
	Excitability of spinal cord neurons	
	Diffuse noxious inhibitory controls (DNIC)	Conditioned pain modulation (CPM)

**Table 2 healthcare-10-01485-t002:** Pressure Pain Threshold (PPT) Values of Neck and Shoulder in Volunteers/Patients with and without Neck/shoulder Pain.

Authors		Object of Study/Subject	Areas of PPT Examination	Device (Units)	PPT Values in Patients/People with Neck and Shoulder Pain; Mean (±SD)	PPT Values in Healthy Volunteers/Control; Mean (±SD)
Arjona Retamal JJ et al.	2021	Chronic neck pain	Upper trapezius	Digital algometer (kg/cm^2^) (FPX 25, Wagner Ins, Greenwich, CT)	Upper trapezius: 1.35–1.56	none
Leon Hernández JV et al.	2021	Chronic neck pain	Upper trapezius	Digital algometer (kg/cm^2^) (Wagner Ins, greenwich, CT, USA)	Uppe tripezius: 4.11–4.14	none
Stieven FF et al.	2021	Chronic neck pain	Upper trapezius	Digital algometer (kgf) (FPX 25, Wagner Ins, Greenwich, CT)	Upper trapezius: 1.35–1.46	none
Oliveira AK et al.	2021	Chronic neck pain	Upper trapezius	Algometer (kg/cm^2^) (PTR-300 model, Instrutherm, São Paulo, SP, Brazil)	Upper trapezius: 1.44–1.76	none
Grimby-Ekman A et al.	2020	Chronic neck-shoulder pain	Upper trapezius	Distal algometer (kpa) (Somedic AB, Farsta, Sweden)	Upper trapezius: 376–411	Upper trapezius: 335–436
Heredia-Rizo AM et al.	2020	Neck-shoulder pain	Upper trapezius	Electronic pressure algometer (kpa) (Somedic AB, Horby, Sweden)	Upper trapezius: 226.1 ± 103.2 (177.8–274.4)	Upper trapezius: 282.2 ± 109.4
Arias-Buria JL et al.	2020	Mechanical neck pain	Triger point area of shoulder	Mechanical algometer (kpa) (Pain Diagnosis and Treatment Inc., New York, NY, USA)	Triger point area of shoulder: 145.4–148.9	none
Shin HJ et al.	2020	Chronic neck pain	Upper trapezius Splenius capitis Levator scapulae	Digital algometer (kg) (Somedic AB, Farsta, Sweden)	Upper trapezius: 2.41–2.56 Splenius capitis: 2.56–2.90 Levator scapulae: 2.07–2.38	none
Rodríguez-Huguet M et al.	2020	Mechanical neck pain	Upper trapezius Suboccipital	Algometer (N/cm^2^) (Wagner Ins, Greenwich, CT, USA)	Upper trapezius: 1.46 Suboccipital: 1.20–1.22	none
Alfawaz SS et al.	2020	Non-specific mechanical neck pain	Upper trapezius	Handheld algometer (N/cm^2^) (Force five™, Wagner Ins, Greenwich, CT, USA)	Upper trapezius: 4.0–4.9	none
Chatchawan U et al.	2019	Chronic tension-type headache (CTTH) Episodic tension-type headaches (ETTH)	Head, neck, shoulder and upper back.	Manual algometer (kg/cm^2^) (Force Dial FDK/FDN Series Mechanical Force Gage; Wagner Ins, Greenwich, CT, USA).	Head, neck, shoulder and upper back: 0.7–1.2	none
Wang-Price S et al.	2018	Neck-shoulder pain	Middle deltoid Levator scapulae Upper trapezius	Handheld computerized pressure algometer (kpa) (Medoc Ltd., Ramat Yishai, Israel)	Middle deltoid: 194.7–228.6 Levator scapulae: 244.1–246.5 Upper trapezius: 167.6–204.1	Middle deltoid: 248.9–309.2 Levator scapulae: 313.7–322.0 Upper trapezius: 229.3–234.6
Murray M et al.	2017	Neck-shoulder pain among military helicopter pilots and crew	Trapezius Upper neck extensors	Handheld electronic pressure algometer (kpa) (Type II Algometer, Somedic Production AB, Sweden)	Trapezius: 405–434 Upper neck extensors: 334–347	none
De Meulemeester KE et al.	2017	Myofascial neck/shoulder pain	Upper and middle trapezius, levator scapulae, infraspinatus and supraspinatus	Wagner FPX Digital Algometer (kg/cm^2^)	Upper and middle trapezius, levator scapulae, infraspinatus and supraspinatus: 16.20–20.63	none
Wassinger CA et al.	2016	Shoulder pain.	Shoulder	Electronic pressure algometer (kg/cm^2^) (Wagner Ins, Greenwich, CT, USA)	Shoulder: 5.67–5.73	none
Toprak Celenay S et al.	2016	Neck pain	C7 and acromion at the middle point of the upper trapezius muscle	Digital algometer (kg/cm^2^) (JTech Medical Industries, ZEVEX Company)	Trapezius: 7.05–7.74	none
Pajediene E et al.	2015	Whiplash associated pain	Upper part of the body	Hand-held pressure algometer (kg/cm^2^) (Pain TestTM Algometer, Wagner Force dial FDK 20)	Upper part of the body: 2.87 ± 1.21	Upper part of the body: 3.71 ± 1.69
Lopez-Lopez A et al.	2015	Chronic neck pain	C2 spinous process	Digital algometer (N/cm^2^) (FDX 25, Wagner Ins, Greenwich, CT, USA)	C2 spinous: 1.49–1.70	none
Ge HY et al.	2014	Computer users with or without pain in the neck-shoulder and forearmregions	Most painful or dominant side of the neck-shoulder region	Digital algometer (kpa) (Somedic AB, Hörby, Sweden)	Most painful or dominant side of the neck-shoulder region: 203.4–308.6	Most painful or dominant side of the neck-shoulder region: 238.7–337.1
Llamas-Ramos R et al.	2014	Chronic neck pain	C7 spinous process	Mechanical algometer (kpa) (Pain Dignosis and Treatment Inc., New York, NY, USA)	C7 spinous process: 188.1 ± 49.4	none
Andersen CH et al.	2014	Chronic neck/shoulder pain	Upper trapezius Lower trapezius	Pressure algometer (kpa) (Algometer Type 2, Somedic, Hörby, Sweden)	Upper trapezius: 277–303 Lower trapezius: 308–383	none
Casanova-Méndez A et al.	2014	Chronic neck pain	Upper tarapezius C4 spinous process T4 spinous process	Analog pressure algometer (kg/cm^2^) (Baseline^®^, FEI Inc., White Plains, NY, USA)	C4 spinous process: 1.96–2.01 T4 spinous process: 3.35–3.70 Upper tripezius: 2.79–3.46	none
Cagnie B et al.	2013	Office workers with mild neck and shoulder complaints	Triger point in Levator scapula/Upper Trapezius/Splenius cervicis	Electronic algometer (N) (compuFET; Hoggan Health Industries, Inc., West Jordan, UT, USA)	Triger point in Levator scapula/Upper Trapezius/Splenius cervicis: 16.2–24.5	none
Yoo IG et al.	2013	Neck-shoulder pain	Upper trapezius Middle trapezius	Dolorimeter pressure algometer (lb) (Fabrication Enterprises, White Plains, NY, USA)	Upper trapezius: 7.2 ± 1.8 Middle trapezius: 5.8 ± 1.4	Upper trapezius: 6.3 ± 2.0 Middle trapezius: 5.0 ± 1.2
Lauche R et al.	2013	Chronic neck pain	Levator scapula Trapezius upper Semispinalis capitis	Digital algometer (kpa) (Somedic AB, Hörby, Sweden)	Levator scapula: 273.4–343.6 Trapezius upper: 229.0–273.7 Semispinalis capitis: 178.9–219.5	none
Casanova-Méndez A et al.	2013	Chronic neck pain	C4 and T4 spinous process Trapezius	Analogue pressure algometer (kg/cm^2^) (Baseline, FEI Inc., White Plains, NY, USA)	C4 spinous process: 1.96–2.01 T4 spinous process: 3.35–3.70 Trapezius: 2.79–3.46	none
Fernández-Pérez AM et al.	2012	Whiplash associated pain	Articular pillar of the C5–6 zygapophyseal joints	Electronic algometer (kpa) (Somedic AB, Sweden)	Articular pillar of the C5–6 zygapophyseal joints: 139.8–158.0	Articular pillar of the C5–6 zygapophyseal joints: 205.4–212.7
Andersen LL et al.	2012	Neck-shoulder pain	Upper Trapezius	Electronic pressure algometer (kpa) (Wagner Ins, greenwich, CT, USA)	Upper trapezius: 219–260	none
Andersen LL et al.	2012	Neck/shoulder pain	Upper trapezius	Electronic pressure algometer (kpa) (Wagner Ins, Greenwich, CT, USA)	Trapezius: 219–260	none
Ge HY et al.	2009	Fibromyalgia syndrome	Upper trapezius	Electronic algometer (kpa) (Somedic AB, Sweden)	Trapezius: 151–156	Trapezius: 151–156
Gerdle B et al.	2008	Whiplash associateddisorders	Trapezius	Electronic pressure algometer (kpa) (Somedic Algometer type 2, Sollentuna, Sweden)	Trapezius, low cervical and spraspinatus: 95–130	Trapezius, low cervical and spraspinatus: 230–300
Lemming D et al.	2007	Whiplash associated pain	Infraspinatus	Electronic algometer (kpa) (Somedic AB, Sweden)	Infraspinatus: 224.6–270.5	none
Ylinen J et al.	2007	Chronic neck pain	Splenius capitis Trapezius Levator scapulae	Hand-held digital pressure algometer (N/cm^2^) (Force fiveTM, Wagner Instruments, Box 1217, Greenwich, CT 06836, USA)	Splenius capitis: 38.9–39.6 Trapezius: 38.3–40.3 Levator scapulae: 60.2–60.7	none
Ylinen J et al.	2007	Chronic neck pain	Strenum Trapezius middle Lavator scapulae Trapezius upper	Handheld algometer (N/cm^2^) (Force five™, Wagner Ins, Greenwich, CT, USA)	Strenum: 31–34 Trapezius middle: 28–31 Lavator scapulae: 43–51 Trapezius upper: 27–31	none
Ojala T et al.	2006	Neck-Shoulder Myofascial Pain Syndrome	Trigger point in neck and shoulder	Dolorimeter (kg/cm^2^) (Pain Diagnostic, Fisher).	Trigger point in neck and shoulder: 5.1–5.3	none
Ylinen J et al.	2005	Chronic neck pain	Sternum Trapezius middle Levator scapulae Trapezius upper	Handheld electronic pressure algometer (N/cm^2^) (Force five™, Wagner Ins, Greenwich, CT, USA)	Strenum: 31–36 Trapezius middle: 28–38 Lavator scapulae: 45–59 Trapezius upper: 28–38	none
Nabeta T et al.	2002	Neck/shoulder pain	Neck Shoulder	Pressure algometer (kg/cm^2^) (Yufu-Seiki, F P Meter, with probe of 10 mm)	Neck: 1.6–1.7 Shoulder: 2.0–2.4	none
Waling K, Sundelin G et al.	2000	Work-related trapezius myalgia	Trapezius upper/middle/lower	Somedicw pressure algometer (kpa) (Somedic Production AB, Sollentuna, Sweden).	Trapezius upper/middle/lower: 194–253	none
Taimela S et al.	2000	Chronic neck pain	Trapezius Levator Scapulae	Mechanical force gauge (N/cm^2^)	Trapezius: 25.9–29.6 Levator Scapulae: 40.9–41.7	none
Otto A et al.	2019	Healthy volunteer	Deltoid Upper trapezius	Pressure algometer (kpa) (Somedic AB, Farsta, Sweden, probe size of 1 cm^2^ surface area)	none	Deltoid: 304.42 ± 103.42 Upper trapezius: 77.3 ± 126.90
Sacramento LS et al.	2017	Healthy Children and Young Adults	Upper Trapezius Supraspinatus Infraspinatus Levator Scapulae Deltoid	Digital pressure algometer (kg/cm^2^) (OE-220, ITO Physiotherapy and Rehabilitation, Ito, Japan)	none	Children: Upper Trapezius: 1.4 ± 0.6 Supraspinatus: 2.0 ± 0.7 Infraspinatus: 2.2 ± 0.7 Deltoid: 2.1 ± 0.8 C5–6 joint: 1.3 ± 0.5 Adults: Upper Trapezius: 2.5 ± 0.9 Supraspinatus: 3.5 ± 1.3 Infraspinatus: 3.8 ± 1.3 Deltoid: 2.7 ± 1.1 C5–6 joint: 2.1 ± 0.7
Wytrążek M et al.	2014	Healthy volunteerwithout triger points	Trapezius (upper part) n = 13 Sternocleido-mastoid n = 12 Deltoid (middle part) n = 48 Infraspinatus n = 13	Algometer (kg/cm^2^) (Force Dial FDK/FDN Series Push Pull Force Gage; Wagner Instruments, Riverside, CT, USA)	none	Trapezius (upper part): 8.38–9.5 Sternocleido-mastoid: 5.92–6.33 Deltoid (middle part): 7.85–8.56 Infraspinatus: 8.32–10
Shin SJ et al.	2012	Healthy worker	Upper trapezius	Dolorimeter (Lb) (Fabrication Enterprises, White Plains, NY, USA)	none	Upper trapezius: 7.3–8.8
Binderup AT et al.	2010	Healthy volunteer	Upper trapezius Middle trapezius Lower trapezius Spinal processes	Hand-held algometer (kpa) (Somedic^®^ Algometer type 2, Sweden)	none	Upper trapezius: 295.2 ± 95.9 Middle trapezius: 347.5 ± 103.5 Lower trapezius: 373.0 ± 121.1 Spinal processes: 369.6 ± 116.5
Saíz-Llamosas JR et al.	2009	Healthy volunteer	C5-C6 zygapophyseal joints	Electronic algometer (kpa) (Somedic AB, Sweden)	none	C5-C6 zygapophyseal joints: 175.4–185.5
Fernández-de-Las-Peñas C et al.	2008	Healthy volunteer	C5-C6 zygapophyseal joints	Electronic algometer (kpa) (Somedic AB, Sweden)	none	C5-C6 zygapophyseal joints: 308.4–334.5
Ge HY et al.	2006	Healthy volunteer	Trapezius Posterolateral neck	Pressure algometer (kpa) (Somedic Algometer type 2, Sollentuna, Sweden)	none	Trapezius: 320–430 Posterolateral neck: 420–445
Ge HY et al.	2005	Healthy volunteer	Trapezius Posterolateral neck	Pressure algometer (kpa) (Somedicw Algometer type 2, Sollentuna, Sweden)	none	Trapezius: 440–550 Posterolateral neck: 365–405
Nie H et al.	2005	Healthy volunteer	Cervical muscle: processus transversus C5 Cervical myotendinous spot: processus transversus C7 Upper trapezius: middle point of processus spinosus C7 and acromion Levator scapulae: 2 cm superior to the angulus superior scapulae Angulus superior scapulae 1 cm medial to the acromioclavicular joint Supraspinatus: 3 cm superior to the middle of spina scapulae Infraspinatus: 3 cm distal to the middle of spina scapulae Middle trapezius: middle point of processus spinosus and medial border of spina scapulae Lower trapezius	Electronic pressure algometer (kpa) (Somedic Algometer type 2, Sweden)	none	Neck/shoulder: Average 322

**Table 3 healthcare-10-01485-t003:** Pressure Pain Threshold (PPT) Values of Low Back in Volunteers/Patients with and without Low Back Pain (LBP).

Authors		Object of Study/Subject	Areas of PPT Examination	Device (Units)	PPT Values in Patients/People with Low Back Pain; Mean (±SD)	PPT Values in Healthy Volunteers/Control; Mean (±SD)
Zywien U et al.	2022	White-collar workers	Lumbar	FDIX RS232 algometer from Wagner (N/cm^2^)	Lumbar: 32.93–64.09	Lumbar: 33.10–64.15
Selva-Sarzo F et al.	2021	Chronic LBP	Lumbar	Wagner Force Dial FDK 20 algometer (kgf)	Lumbar: 3.68–6.83	none
Dias LV et al.	2021	Chronic LBP	Bilaterally 5 cm from the spinal process of L3 and L5	Pressure algometer (kgf) (EMG System^®^ of Brazil).	L3: 3.5–5.2 L5: 3.6–5.3	none
Nim CG et al.	2021	Non-specific LBP	L1 segment L2 segment L3 segment L4 segment L5 segment	Pressure algometer (kpa) (Model 2, Somedic, Sweden)	L1 segment: 522 ± 244 L2 segment: 482 ± 228 L3 segment: 472 ± 225 L4 segment: 455 ± 225 L5 segment: 445 ± 229	none
Mailloux et al.	2021	Healthy volunteers	Lumbar erector spinae (LES) 2–3 cm laterally to L4/L5 S1 spinous process	Handheld digital algometer (kpa) (1-cm^2^ probe–FPIX, Wagner Instruments, Greenwich, CT, USA)	none	LES: 547.2–559.7 S1 spinous process: 517.2–536.7
Leemans L et al.	2020	Chronic LBP	2 cm lateral to the L3 spinous process 2 cm lateral to the L5 spinous process Near the poste-rior superior iliac spines (PSIS)	Digital pres-sure algometer (kgf) (Wagner Force Ten)	Lower back: 6.7–7.0	none
Nim CG et al.	2020	Non-specific LBP	Lumbar	Pressure algometer (kpa) (Model 2, Somedic, Sweden)	Lumbar (with pain): 488.73 ± 330.95 Lumbar (with stiffness): 436.6 ± 364.9	none
Volpato MP et al.	2020	Chronic non-specific LBP	BL23 (Shenshu) BL24 (Qihaishu) BL25 (Dachangshu)	Pressure algometer (unknown) (EMG 830C, EMG System, São José dos Campos, Brazil)	BL23 (Shenshu): 7025–7844 BL24 (Qihaishu): 7258–7285 BL25 (Dachangshu): 6445–7173	none
Wang-Price S et al.	2020	LBP	Lumbar	Hand-held computerized pressure algometer (kpa) (Medoc Ltd., Ramat Yishai, Israel)	Lumbar: 383.4 ± 185.5	none
Vaegter HB et al.	2020	LBP	Erector spinae muscle	Manual pressure algometry (kpa) (Somedic Sales AB)	Erector spinae muscle: 450–586	none
Fagundes Loss J et al.	2020	LBP	Spinal Process/Erector	10 kgf analogic pressure algometer (kgf) (Wagner Instruments, Greenwich, CT-USA)	Spinal Process: 6.1–7.0 Erector: 6.8–7.2	none
Plaza-Manzano G et al.	2020	Lumbar radiculopathy	Common peroneal Tibialis	Mechanical pressure algometer (kg/cm^2^) (Pain Diagnosis and Treatment Inc., New York)	Common peroneal: 2.1–2.3 Tibialis: 3.2–3.4	none
Moreira RFC et al.	2020	Hospital nursing assistants with LBP	Dorsal longissimus	Hand-held algometer (kgf/cm^2^) (Pain Diagnostic Treatment, New York, USA)	Dorsal longissimus: 5.52–6.55	none
Bond BM et al.	2020	Non-specific LBP	Lumbar Paraspinal Musculature	Digital algometer (kg/cm^2^) (Wagner Instruments, Greenwich, Connecticut)	Lumbar Paraspinal Musculature: 3.36–3.39	none
Chapman KB et al.	2020	Chronic LBP	The most painful side of the back	Pressure algometer (N/cm^2^) (Wagner Instruments, Greenwich, CT, USA)	The most painful side of the back: 28.7 ± 4.1	none
Aspinall SL et al.	2020	Chronic LBP	Lumber	Digital pressure algometer (kg/cm^2^) (FPIX 50, Wagner Instruments, Connecticut, USA)	Lumber: 4.1–4.4	Lumber: 5.5 ± 4.1
Aspinall SL et al.	2020	LBP	Lumber	Digital pressure algometer (kg/cm^2^) (FPIX 50, Wagner Instruments, Connecticut, USA)	Lumber: 4.14–4.30	none
Petersson M et al.	2020	Healthy volunteers	Spinous processes from L1 to L5 (L1-L2, L2-L3, L3-L4 and L4-L5).	Pressure algometer (kpa) (SOMEDIC Electronics brand, Solna, Sweden)	none	Spinous processes L1-L2: 353–405 L2-L3: 319–361 L3-L4: 299–370 L4-L5: 306–321
Aspinall SL et al.	2019	LBP	Lumber	Digital pressure algometer (kg/cm^2^) (FPIX 50, Wagner Instruments, Connecticut, USA)	Lumber: 5.3 ± 3.3	none
Alfieri FM et al.	2019	Non-specific LBP	ParavertebralL4–5	J Tech algometer (lb) (Salt Lake City, UT, USA)	Paravertebral: 8.1–8.3 L4–5: 8.3 ± 3.6	Paravertebral: 12.9–13.1 L4–5: 12.8 ± 5.6
McPhee ME et al.	2019	Recurrent LBP	L1 and L5 (3.5 cm lateral to the L1 and L5 spinous processes)	Rubbertipped handheld pressure algometer (kpa) (Somedic, Norra Mellby, Sweden)	L1: 405 L5: 415	L1: 550 L5: 560
Kołcz A et al.	2019	Professionally active nurses	Erector spinae muscle	AlgoMed FPIX 50 (Medoc, Yishai, Israel)	none	Erector spinae muscle: 30 ± 1.8
Marcuzzi A et al.	2018	Acute LBP	Back	Pressure algometer (kpa) (FDK40; Wagner Instrument, Greenwich, CT)	Back: 2.6–2.7	Back: 2.2 ± 0.3
de Carvalho RC et al.	2018	LBP	2 cm lateral to the L1, 3, and 5 spinous process	Pressure algometer (unknown) (EMG 830C, EMG System, São José dos Campos, Brazil)	2 cm lateral to the L1, 3, and 5 spinous process: 5.29–5.68	none
Bodes Pardo G et al.	2018	Chronic LBP	Spinal process of L3	Fisher algometer (kg/cm^2^) (Force Dial model FDK 40)	Spinal process of L3: 2.8–3.0	none
Joseph LH et al.	2018	Elite female weight lifters with Chronic non-specific LBP	Lumbar region	Pressure algometer (kpa) (Algometer type II, Somedic SenseLab AB, Sweden)	Lumbar region: 472.28–440.64	none
O’Neill S et al.	2018	Non-specific LBP	Back-Lumbar	Pressure algometer (kpa) (Somedic model 2, 1 cm^2^ probe, Hørby, Sweden),	Back-Lumbar: 322 ± 196	Back-Lumbar: 398 ± 194
Yildiz SH et al.	2017	Chronic LBP	L1 Paravertebral level L3 Paravertebral L5 Paravertebral	Manual algometer (unknown)	L1 Paravertebral: 6.7–7.3 L3 Paravertebral: 6.8–7.3 L5 Paravertebral: 6.7–7.2	L1 Paravertebral: 7.6–9.5 L3 Paravertebral: 7.4–9.4 L5 Paravertebral: 7.6–9.5
Shane LG et al.	2017	LBP	L3, L4, and L5 paraspinal muscles	Digital pressurealgometer (N/cm^2^) (Wagner Force Ten FDX, Wagner Instruments, Greenwich, CT)	L3, L4, and L5 paraspinal muscles: 6.32–6.59	none
Paungmali A et al.	2017	Chronic non-specific LBP	Lumbar region	Pressure algometer (kpa) (Algometer type II; Somedic Production AB, Sollentuna, Sweden)	Lumbar region: 509.8 ± 133.3	none
Mohanty PP et al.	2017	Coccydynia	Coccygeal region	Modified syringe algometer (unknown)	Coccygeal region: 2.2–2.5	none
Calvo-Lobo C et al.	2017	Myofascial pain syndrome	Lumbar erector spinae muscles	Manual mechanical algometer (kg/cm^2^) (FDK/FDN, Wagner Instruments, 1217 Greenwich, CT 06836)	N = 20 (Active trigger points): 2.97 ± 0.82 N = 20 (Latent trigger points): 3.56 ± 0.77	N = 20 (Controls points): 4.49 ± 0.90
Nothnagel H et al.	2017	Healthy volunteers	Lumbar paraspinal	Pressure gauge device (kpa) (FDN200, Wagner Instruments, Greenwich, CT, USA)	none	Lumbar paraspinal: 589.68–628.32
Farasyn A et al.	2016	Chronic non-specific LBP	Erector spinae T8 Erector spinae T10 Erector spinae L1 Erector spinae L3 Gluteus maximus pars superior Gluteus maximus pars inferior	Electric pressure algometer (kg/cm^2^) (MYOMETER, Penny & Giles, U.K.),	Erector spinae T8: 3.96 ± 1.30 Erector spinae T10: 3.73 ± 1.10 Erector spinae L1: 3.71 ± 1.20 Erector spinae L3: 5.29 ± 1.27 Gluteus maximus pars superior: 3.73 ± 1.17 Gluteus maximus pars inferior: 3.84 ± 0.94	Erector spinae T8: 7.03 ± 1.50 Erector spinae T10: 7.77 ± 1.31 Erector spinae L1: 8.69 ± 1.66 Erector spinae L3: 9.86 ± 1.41 Gluteus maximus pars superior: 9.10 ± 1.83Gluteus maximus pars inferior: 8.81 ± 2.01
Imamura M et al.	2016	Chronic non-specific LBP	Gluteus medius middle portion Gluteus medius posterior portion Gluteus minimus Gluteus maximus Piriformis Quadratus Lumborum Iliopsoas Ligaments T12-L1/ L1-L2/ L2-L3/L3-L4/L4-L5 L5-S1/S1-S2/S2-S3	Pressure algometer (kg/cm^2^) (Pain Diagnostics, Great Neck, NY).	Gluteus medius middle portion: 5.08 ± 2.12 Gluteus medius posterior portion: 4.92 ± 1.95 Gluteus minimus: 5.36 ± 2.20 Gluteus maximus: 5.20 ± 2.25 Piriformis: 5.70 ± 2.60 Quadratus Lumborum: 4.61 ± 1.85 Iliopsoas: 3.97 ± 1.78 Ligaments T12-L1: 5.09 ± 2.63 L1-L2: 4.97 ± 2.71 L2-L3: 4.96 ± 2.73 L3-L4: 4.93 ± 2.58 L4-L5: 4.74 ± 2.22 L5-S1: 4.83 ± 2.54 S1-S2: 5.00 ± 2.91 S2-S3: 5.25 ± 2.91	none
Balaguier R et al.	2016	Healthy volunteers	Lumbar spinal processes L1-L5	Somedic Algometer (kpa) (Type 2, Sollentuna, Sweden)	none	Lumbar spinal processes L1: 593.4–654.6 L2: 616.0–664.1 L3: 573.4–638.6 L4: 560.9–657.6 L5: 607.5–653.9
Weinkauf B et al.	2015	Healthy volunteers	Lumbar	Homogeneous pressure (kpa) (Wagner Instruments, USA)	none	Lumbar: 703 ± 24
Falla D et al.	2014	Chronic non-specific LBP	8 locations at lumbar	Electronic algometer (kpa) (Somedic Production, Stockholm, Sweden)	8 locations at lumbar: 268.0 ± 165.9	8 locations at lumbar: 320.1 ± 162.1
Imamura M et al.	2013	Chronic LBP	Suprainterspinous ligaments situated between T12–L/L1–L2/L2–L3/L3–L4/L4–L5/L5–S1/S1–S2/S2–S3	Pressure algometer (kg/cm^2^) (Pain Diagnostics, Great Neck, NY)	Suprainterspinous ligaments situated between T12–L: 5.06 ± 2.47 L1–L2: 4.84 ± 2.23 L2–L3: 4.64 ± 2.05 L3–L4: 4.83 ± 2.19 L4–L5: 4.65 ± 1.74 L5–S1: 5.40 ± 3.20 S1–S2: 5.77 ± 3.55 S2–S3: 5.71 ± 3.51	Suprainterspinous ligaments situated between T12–L: 7.16 ± 2.53 L1–L2: 7.29 ± 2.21 L2–L3: 7.49 ± 2.01 L3–L4: 7.46 ± 2.57 L4–L5: 8.10 ± 2.46 L5–S1: 8.38 ± 2.40 S1–S2: 8.89 ± 2.63 S2–S3: 8.75 ± 2.18
de Oliveira RF et al.	2013	Chronic non-specific LBP	L3/L5 spinous process	Pressure algometer (N) (Kratos model DDK, Kratos Ltd., São Paulo, Brazil)	L3/L5 spinous process: 48.90–49.63	none
Neziri AY et al.	2012	Chronic LBP	Site of most severe pain at low back Nonpainful site at low back	Electronic pressure algometer (kpa) (Somedic, Hörby, Sweden)	Site of most severe pain at low back: 168 ± 113 Nonpainful site at low back: 249 ± 132	Site of most severe pain at low back: 352 ± 131 Nonpainful site at low back: 352 ± 131
Zheng Z et al.	2012	Chronic non-specific LBP	Lumbar tender point	Model OE-220, made in Ito ultrashort wave corporation of Japan (kg/cm^2^)	Lumbar tender point: 3.7–3.8	none
McSweeney TP et al.	2012	Healthy volunteers	Paravertebral soft tissue at L1	Handheld manual digital pressure algometer (N) (Wagner FPX 25)	none	Paravertebral soft tissue at L1: 53.7–60.1
Yu X et al.	2012	Healthy volunteers	L5-S1 zygapophyseal joints	Mechanical pressure algometer (kg/cm^2^) (Wagner, Greenwich, CT)	none	L5-S1 zygapophyseal joints: 4.87–5.01
O’Neill S et al.	2011	Chronic LBP	Spinous process of L4	Pressure algometer (kpa) (Model 2, Somedic, Sweden)	Spinous process of L4: 677	Spinous process of L4: 755
Binderup AT et al.	2011	Healthy volunteers (cleaners)	Spinal processes L1–5 Low back Erector Spinae	Hand-held pressure algometer (kpa) (Somedic Algometer type 2, Sweden)	none	Spinal processes L1–5: 427.9 ± 204.7 Erector Spinae: 409.8 ± 194.5
Hirayama J et al.	2006	Lumbar disc herniation	2 cm lateral to the L1, 3 and 5 spinous process 5 cm lateral to the L1 and 3 spinous process	Electronic pressure algometer (kpa) (Somedic, Farsta, Sweden)	2 cm lateral to the L1: 451.4 ± 214.8 L3: 348.5 ± 179.6 L5: 267.7 ± 105.1 5 cm lateral to the L1: 325.6 ± 133.0 L3: 297.1 ± 157.2	2 cm lateral to the L1: 438.1–441.3 L3: 418.5–424.5 L5: 395.5–401.3 5 cm lateral to the L1: 331.8–362.1 L3: 314.0–335.3
Farasyn A et al.	2005	Subacute non-specific LBP	Erector spinae mass T6, T10, L1, L3 and L5 Gluteus maximus Gluteus medius Tensor fasciae latae (TFL).	Mechanical Fischer pressure algometer (kg/cm^2^) (Pain Diagnostics and Thermography, Great Neck, NY, USA)	Erector spinae mass T6: 6.7 ± 1.8 T10: 6.6 ± 1.1 L1: 6.4 ± 1.2 L3: 5.3 ± 14 L5: 7.2 ± 1.6 Gluteus maximus: 6.4 ± 1.6 Gluteus medius: 6.1 ± 1.6 TFL: 6.3 ± 1.5	Erector spinae mass T6: 7.6 ± 1.1 T10: 7.4 ± 1.1 L1: 7.4 ± 1.2 L3: 7.7 ± 1.7 L5: 9.5 ± 1.2 Gluteus maximus: 8.0 ± 1.5 Gluteus medius: 7.2 ± 1.5 TFL: 7.1 ± 1.4

## Data Availability

Not applicable.

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
