# Peer review of "Current Concept of Quantitative Sensory Testing and Pressure Pain Threshold in Neck/Shoulder and Low Back Pain"

_healthcare, 2022, doi:10.3390/healthcare10081485_

Round 1
Reviewer 1 Report
Thank you for letting me review this interesting manuscript. In general, I would like to say that is an interesting paper, that gives quite important information to the readers. However, I have few concerns about how the manuscript is designed and written.
- Table 1. Please delete de references column and add the references in the paragraph.
- Line 100. Please do not use rethorical questions. Rewrite the subtitle.
- Line 130. Plese avoid first person writting and the use of subjective and personal statements. Rewrite the paragraph using objective sentences and based on the scientific evidence.
- Revise the entire manuscrip trying to avoid the subjective written.
- I would like to say to the authors, that I really appreciate the effort that all of them have made in searching, reading, and evaluating all the studies included in the Tablles.
Hower, there is no description about the type of the study, there is no registration of the review, there is no material and methods explanation... and there is no synthesis of results, explaining the main results in summary.
In addition, tables are extremely huge, so the authros shoudl find the way to improve the readability of the tables.
Author Response
Reviewer 1
Thank you for letting me review this interesting manuscript. In general, I would like to say that is an interesting paper, that gives quite important information to the readers. However, I have few concerns about how the manuscript is designed and written.
Dear reviewer 1
We are grateful for the critical review of our manuscript with kind offer of invaluable comments to improve it. I rewrite all the points you pointed out. Please see below the responses to all your comments.
- Table 1. Please delete de references column and add the references in the paragraph.
Responses: We appreciate for raising the Table 1 issues. We delete the references column and add the references in the paragraph.
- Line 100. Please do not use rhetorical questions. Rewrite the subtitle.
Responses: Thanks again. We rewrite the subtitle.
- Line 130. Please avoid first person writing and the use of subjective and personal statements. Rewrite the paragraph using objective sentences and based on the scientific evidence.
- Revise the entire manuscript trying to avoid the subjective written.
Responses: We appreciate for raising the important issue in my manuscript. We delete all my subjective and personal statements; in addition, we added the references number in my manuscript.
- I would like to say to the authors, that I really appreciate the effort that all of them have made in searching, reading, and evaluating all the studies included in the Tables.
However, there is no description about the type of the study, there is no registration of the review, there is no material and methods explanation... and there is no synthesis of results, explaining the main results in summary. In addition, tables are extremely huge, so the authors should find the way to improve the readability of the tables.
Responses: We appreciate for raising the important issue in my manuscript. I added the type of the study and the flow chart of inclusion criteria in Figure 1 and in the session of 4.1-4.2 in pp4-5.
We simplified the Tables. Are they readable for readers? If not so, we modified again following your suggestions.
In addition, we explained the data summary and discussed the data the previous issues following all of the articles review in the discussion’s session (line 295-309). We described the meaning of this review manuscript in pain research in discussions session (310-312)

Reviewer 2 Report
First of all, congratulations to the authors for the choice of such a relevant topic.
I will now list some aspects that need to be resolved and addressed for possible publication:
1. The manuscript does not follow the typical structure of research articles: Introduction, Material and Methods....
2. The manuscript should be a systematic review (with its typical structure), specifying the inclusion and exclusion criteria. Likewise, the review should be registered in PROSPERO.
3. The discussion and conclusions section is very simple for such a complex subject.
4. The limitations of the study should be indicated.
Author Response
Reviewer 2
First of all, congratulations to the authors for the choice of such a relevant topic.
I will now list some aspects that need to be resolved and addressed for possible publication:
Dear reviewer 2
We are grateful for the critical review of our manuscript with kind offer of invaluable comments to improve it. I rewrite all the points you pointed out. Please see below the responses to all your comments.
- The manuscript does not follow the typical structure of research articles: Introduction, Material and Methods....
Responses: We appreciate for raising the important issue in my manuscript. This is the review articles; therefore, we wrote, Introduction, QST and PPT explanation at first. After that, in the session 4, we just started methods of literature search and inclusion criteria. We added the flowchart of references—screening process in this systematic review in Figure 1. Thank you very much for letting me know the issue.
- The manuscript should be a systematic review (with its typical structure), specifying the inclusion and exclusion criteria. Likewise, the review should be registered in PROSPERO.
Responses: Thank you again for raising the important issue in my manuscript. I added the type of the study and the flow chart of inclusion criteria in Figure 1 and in the session of 4.1-4.2 in pp4-5. We conducted a systematic review according to PRISMA (Preferred Reporting Items for Systematic Re-views and Meta-Analyses) guidelines (https://prisma-statement.org/).
- The discussion and conclusions section is very simple for such a complex subject.
- The limitations of the study should be indicated.
Responses: We appreciate for raising the important issue in my manuscript. We wrote down the discussions and conclusions depend on the review articles (line 295-311). The aim of the review articles is to reveal and summarize the normative and abnormal PPT score from the previous papers because these kinds of values are necessary for widely use of PPT in general clinics to assess the chronic pain patients. In these reasons, the results and conclusion are simple one. Even if the results were simple one, we understood that we still have the big issues on PPT analysis. It is not the limitation; however, we described the issues on PPT examination in the conclusion session.

Reviewer 3 Report
The work engaged the fully literature review regarding the quantitative sensory testing (QST) and pressure pain threshold (PPT) for examining and analyzing the neck/shoulder and low back pain. In my opinion, research on identifying the pain threshold should be encouraged and appreciated, especially the literature review and discussion conducted like this study. Any useful findings or conclusions drawn are important foundations for future investigation. Therefore, I may recommend this review paper for publication to Healthcare. Even so, I still have some revisions and suggestions for this manuscript as follows.
1. The literature review process should basically include the definition, inclusion/exclusion criteria, as well as the final discussion and suggestions. This paper seems not have a complete description in this part, but directly shows the results, which may affect the referencing for the follow-up research. Thus, I suggest that a figure should be added to present the flow diagram of the identification, screening, eligibility, and inclusion of the studies in this systematic review
2. The abstract should indicate the original number of articles selected for the review process, then how many key studies were finally analyzed, which is especially important for a literature review study.
3. As stated by authors, there have been no reliable references for PPT values of neck/shoulder and back pain because the data vary depending on the devices used, the measurement units, and the area examined. This may be the key issue to challenge the authors how to summarize the results and propose some important findings from their reviews. Based on these findings and suggestions, the future studies can be developed and avoid fumbling without direction.
4. It is suggested here that the Discussion and Conclusions should be divided into two separate sections, and the conclusions should be described as brief as possible.
5. The tables are hard to be read. I suggest further editing.
Author Response
Reviewer 3
The work engaged the fully literature review regarding the quantitative sensory testing (QST) and pressure pain threshold (PPT) for examining and analyzing the neck/shoulder and low back pain. In my opinion, research on identifying the pain threshold should be encouraged and appreciated, especially the literature review and discussion conducted like this study. Any useful findings or conclusions drawn are important foundations for future investigation. Therefore, I may recommend this review paper for publication to Healthcare. Even so, I still have some revisions and suggestions for this manuscript as follows.
Dear reviewer 3
We are grateful for the critical review of our manuscript with kind offer of invaluable comments to improve it. I rewrite all the points you pointed out. Please see below the responses to all your comments.
- The literature review process should basically include the definition, inclusion/exclusion criteria, as well as the final discussion and suggestions. This paper seems not have a complete description in this part, but directly shows the results, which may affect the referencing for the follow-up research. Thus, I suggest that a figure should be added to present the flow diagram of the identification, screening, eligibility, and inclusion of the studies in this systematic review
Responses: We appreciate for raising the important issue in my manuscript. I added the type of the study and the flow chart of inclusion criteria in Figure 1 and added the methods in the session of 4.1-4.2 in pp4-5.
- The abstract should indicate the original number of articles selected for the review process, then how many key studies were finally analyzed, which is especially important for a literature review study.
Responses: Thank you again for kind your suggestion. We added the original number of articles selected for the review process in abstract.
- As stated by authors, there have been no reliable references for PPT values of neck/shoulder and back pain because the data vary depending on the devices used, the measurement units, and the area examined. This may be the key issue to challenge the authors how to summarize the results and propose some important findings from their reviews. Based on these findings and suggestions, the future studies can be developed and avoid fumbling without direction.
Responses: We appreciate for raising the important issue in my manuscript. In the discussion session (line 315-319) and conclusion session (line 326-331), we added the key issues on PPT examination and described the need of worldwide standardized guideline on PPT.
- It is suggested here that the Discussion and Conclusions should be divided into two separate sections, and the conclusions should be described as brief as possible.
Responses: Thank you again for kind your suggestion. We separated the conclusions session.
- The tables are hard to be read. I suggest further editing.
Responses: Thank you again for kind your suggestion. We simplified the Tables. Are they readable for readers? If not so, we modified again following your suggestions.
I thought that readers would like to know the large information from the Table, like looking up the dictionary.

Round 2
Reviewer 1 Report
The authors have considered all the suggestions and commentas I have made. I just have few minor comments:
- "The PPT is one of the solutions to examining patients with chronic pain for digitalization". It is not a solution is an instrument. Please be careful with the statements.
- "To achieve wide and general use of PPT, we think that all articles on PPT related to neck/shoulder and LBP in volunteers/patients with and without chronic pain need to be reviewed". Please avoid first person use.
- Please, revise the entire manuscript taking into consideration these comments.
- the references column in the tables can be eliminated or the citation placed next to the author to reduce its length.
-The units column can be removed and included within the device column
Author Response
Dear reviewer 1
We are grateful for the critical review of our manuscript with kind offer of invaluable comments to improve it. I rewrite all the points you pointed out. Please see below the responses to all your comments.
The authors have considered all the suggestions and commentas I have made. I just have few minor comments:
- "The PPT is one of the solutions to examining patients with chronic pain for digitalization". It is not a solution is an instrument. Please be careful with the statements.
Responses: We appreciate for raising my mistake in my manuscript. We change the below sentences in my manuscript.
- "The PPT examination is one of the solutions to examining patients with chronic pain for digitalization".
- "To achieve wide and general use of PPT, we think that all articles on PPT related to neck/shoulder and LBP in volunteers/patients with and without chronic pain need to be reviewed". Please avoid first person use.
- Please, revise the entire manuscript taking into consideration these comments.
Responses: We appreciate for raising my mistake in my manuscript. We change the below sentences in my manuscript.
It is necessary to be reviewed all articles on PPT related to neck/shoulder and LBP in volunteers/patients with and without chronic pain to achieve wide and general use of PPT examination.
- the references column in the tables can be eliminated or the citation placed next to the author to reduce its length.
-The units column can be removed and included within the device column
Responses: We appreciate for your valuable suggestions. We eliminated the references in Tables. We included the units within the device column.

Reviewer 2 Report
No comments
Author Response
Dear reviewer 2
We are grateful for the critical review of our manuscript with kind offer of invaluable comments to improve it.
Thank you very much again. Best regards, Hidenori
Reviewer 3 Report
I am pleased to say that the author has made revisions based on my suggestions and questions. I originally encouraged the publication of such articles (review papers), which can provide readers or researchers in the related field with a good studying guide. In addition, Figure 1 is pretty good to indicate how the papers were screened and the results. As for the last question about the readability of the tables in the first review, I think there may be a trade-off. I agree with the authors that readers would like to know the large information from the tables, like looking up the dictionary, if the readability of the tables is sufficient. In terms of providing more information, I agree with the author on this point
Author Response
Dear reviewer 3
We are grateful for the critical review of our manuscript with kind offer of invaluable comments to improve it.
Thank you very much again. Best regards, Hidenori